# Molecular Effects of Silver Nanoparticles on Monogenean Parasites: Lessons from *Caenorhabditis elegans*

**DOI:** 10.3390/ijms21165889

**Published:** 2020-08-16

**Authors:** Citlalic A. Pimentel-Acosta, Jorge Ramírez-Salcedo, Francisco Neptalí Morales-Serna, Emma J. Fajer-Ávila, Cristina Chávez-Sánchez, Humberto H. Lara, Alejandra García-Gasca

**Affiliations:** 1Centro de Investigación en Alimentación y Desarrollo, Unidad Mazatlán en Acuicultura y Manejo Ambiental, Mazatlán, Sinaloa 82112, Mexico; citlalic.pimentel@estudiantes.ciad.mx (C.A.P.-A.); francisco.morales@ciad.mx (F.N.M.-S.); efajer@ciad.mx (E.J.F.-Á.); marcris@ciad.mx (C.C.-S.); 2Unidad de Microarreglos, Instituto de Fisiología Celular, Universidad Nacional Autónoma de México, Avenida Universidad 3000, Mexico City 04510, Mexico; jramirez@ifc.unam.mx; 3CONACYT, Centro de Investigación en Alimentación y Desarrollo, Unidad Mazatlán en Acuicultura y Manejo Ambiental, Mazatlán, Sinaloa 82112, Mexico; 4Department of Biology and South Texas Center for Emerging Infectious Diseases, The University of Texas at San Antonio, San Antonio, TX 78249, USA; humberto.laravillegas@utsa.edu

**Keywords:** silver nanoparticles, monogenean parasites, gene expression, microarray analysis, *Caenorhabditis elegans*, *Cichlidogyrus*

## Abstract

The mechanisms of action of silver nanoparticles (AgNPs) in monogenean parasites of the genus *Cichlidogyrus* were investigated through a microarray hybridization approach using genomic information from the nematode *Caenorhabditis elegans*. The effects of two concentrations of AgNPs were explored, low (6 µg/L Ag) and high (36 µg/L Ag). Microarray analysis revealed that both concentrations of AgNPs activated similar biological processes, although by different mechanisms. Expression profiles included genes involved in detoxification, neurotoxicity, modulation of cell signaling, reproduction, embryonic development, and tegument organization as the main biological processes dysregulated by AgNPs. Two important processes (DNA damage and cell death) were mostly activated in parasites exposed to the lower concentration of AgNPs. To our knowledge, this is the first study providing information on the sub-cellular and molecular effects of exposure to AgNPs in metazoan parasites of fish.

## 1. Introduction

Silver nanoparticles (AgNPs) have shown high antimicrobial activity; several studies have demonstrated the potential of AgNPs to eliminate pathogens like bacteria, fungi, viruses and parasites affecting humans [1,2,3,4,5,6]. For this reason, the effectiveness of AgNPs as antimicrobials in controlling pathogens in aquaculture has recently been investigated, showing that AgNPs can eliminate bacteria [7,8] viruses [9,10] and parasites [11,12] in fish and aquatic invertebrates. Thus, AgNPs may be used as a potential therapeutic agent to control pathogens in aquaculture [13,14].

Monogeneans (Platyhelminthes) are ectoparasites commonly found on the gills of marine and freshwater fish. Some species may cause disease outbreaks and mortality of farmed fish, and the parasitic infection is difficult to control. Some treatments, such as formalin, salt, and some drugs such as praziquantel, are partially effective because they do not eliminate all the stages of the parasite [15,16,17].

Recently, our research team discovered that AgNPs could eliminate monogenean parasites of the genus *Cichlidogyrus* [12]. This is important because *Cichlidogyrus* parasites are commonly found in farmed tilapia around the world. The toxicity of AgNPs depends on several factors, such as nanoparticle size (smaller and uncoated nanoparticles are usually more toxic), the concentration of silver ions (AgNPs are dose-dependent) [18] and the sensitivity of the organism exposed [19,20,21,22,23]. For these reasons, the mechanisms of action can be varied. Some studies have identified biochemical and molecular responses caused by exposure to AgNPs; the best characterized is the antimicrobial action of AgNPs on bacteria and fungi, and these mechanisms include adhesion of AgNPs to the cell surface, internalization and damage of biomolecules and organelles, damage to cellular structures generating reactive oxygen species (ROS) and free radicals causing oxidative stress and toxicity, and modulation of signal transduction pathways [24,25,26].

Few studies have evaluated the toxicity and mechanisms of action of AgNPs in worms; most of these studies have been carried out on the model organism *Caenorhabditis elegans*, and are practically inexistent in monogenean parasites. Changes in transcription profiles after AgNP exposure have been described by transcriptomic approaches, showing that AgNPs induce mitochondrial membrane damage; metal detoxification responses endocytosis and lysosomal function, increased ROS formation (and thus oxidative stress), changes in signal transduction pathways like mitogen-activated protein kinase PMK-1/p38 pathway, DNA damage, and apoptosis [27,28,29,30,31,32,33,34,35]—factors that all affect the lifespan of parasites. Nevertheless, the effects caused by AgNPs can vary depending on the type and concentration of AgNPs. Some authors have suggested that the effects may be due to the synergy of AgNPs and the released silver ions and cause cytotoxicity in a dose-dependent manner [32]. In *C. elegans,* some transcriptomic effects of AgNPs were similar to silver ions (AgNO_3_), indicating that both can alter gene expression profiles in pathways associated with stress response, reproduction an lysosomal activity; these results suggest that the effects are partially due to dissolved silver ions [32,35].

To date, it is known that AgNPs can eliminate monogenean parasites and cause damage on the tegument; we have previously described that parasites exposed to AgNPs at 6 μg/L for 1 h show formation of vacuoles and slight swelling of the tegument, while parasites exposed to 36 μg/L showed swelling, loss of corrugations, and disruption of the tegument [12]. Thus, in this study, the molecular mechanisms of action of AgNPs in monogenean parasites of the genus *Cichlidogyrus* were investigated through a *C. elegans* microarray hybridization approach. 

## 2. Results

### 2.1. Heterologous Microarray Hybridization

Microarray analysis revealed the hybridization of *Cichlidogyrus* genes that are compatible with *C. elegans* genes. Microarray technology has been useful in the study of gene expression to identify mechanisms of disease and accelerate drug discovery, however, gene expression and mechanistic studies in non-model species remain a challenge because of the precarious genomic information available. Here, we decided to use a *C. elegans* microarray to determine gene expression profiles in *Cichlidogyrus* monogenean parasites exposed to AgNPs, considering that genomic information in *Cichlidogyrus* species is scarce. The results showed the hybridization of 19,539 genes with the lower AgNP concentration (6 µg/L), and 19,472 genes with the higher concentration (36 µg/L), out of 20,000 genes contained in the *C. elegans* microarray (Figure 1, Appendix A). We used heterologous microarray hybridization (instead of New Generation Sequencing (NGS)-based technologies such as RNA-Seq) because it represented an interesting and inexpensive strategy; the microarray for *C. elegans* was available, and this species is phylogenetically related to the target species [36,37,38]. This approach provided sufficient information to determine the main mechanisms of action of AgNPs in monogenean parasites. 

### 2.2. Differential Gene Expression in Parasites Exposed to AgNPs

A ≥2 z-score (*p* ≤ 0.05) boundary in gene expression was set to identify significantly differentially up- and down-regulated genes. Gene expression profiles (≥2 z-score, *p* ≤ 0.05) in parasites showed some differences depending on the concentration of AgNPs (6 or 36 µg/L) to which they were exposed. Parasites exposed to 6 µg/L of AgNPs displayed 424 and 328 up- and down-regulated genes, respectively, whereas parasites exposed to 36 µg/L displayed 422 and 387 up- and down-regulated genes, respectively (Figure 2).

Differential gene expression was analyzed with the Bioinformatics Resources DAVID 6.8 to identify molecular pathways affected by AgNP exposure in monogenean parasites (Figure 3). Interestingly, both concentrations of AgNPs seemed to activate similar biological processes, although gene expression profiles were different (see below).

### 2.3. Dysregulated Biological Processes and Molecular Pathways

The top 30 up- and down-regulated genes in parasites exposed to 6 µg/L AgNPs are shown in Table 1 and Figure 4A. These genes are involved in different biological processes such as detoxification (*mrp-4*, *nhr-49*), neurotoxicity (*zig-5*, *zag-1*, *ahr-1*, *cfz-2*, *unc-11*, *fat-3*, *unc-26*), modulation of cell signaling (*akt-2, pqm-1, pdl-1*), cell death (*pqn-41, rfc-2, hsr-9*), reproduction and embryonic development (*lin-9*, *syp-3*, *lap-1*, *hmr-1*, *cyb-2.1*, *spe-11*, *nhr-23*, *spe-29*, *rrf-3*, *nhr-40*, *hpl-2*, *pat-6*), and tegument organization (*ifb-1*, *cut-1*).

In the same way, the top 30 up- and down-regulated genes in parasites exposed to 36 µg/L AgNPs compose a different set of genes (Table 2, Figure 4B), but are involved in the same biological processes of detoxification (*pcs-1*), neurotoxicity (*str-2*, *rab-3*, *zag-1*, *smp-1*, *ldb-1*, *ehs-1*), modulation of cell signaling (*akt-1*, *vhp-1*, *jkk-1*), DNA damage (*ceh-30*) reproduction and embryonic development (*pal-1*, *goa-1*, *cdc-25.1*, *ptl-1*, *lag-1*, *tra-2*, *nhr-40*, *msi-1*, *lev-11*, *efn-2*) and tegument organization (*pat-6*, *ajm-1*, *lon-3*, *cut-1*). 

These results indicate that both concentrations induce the similar biological processes in *Cichlidogyrus* parasites through different mechanisms.

## 3. Discussion

In a previous report, the efficacy of AgNPs against *Cichlidogyrus* parasites was demonstrated [12]. Other studies have shown that AgNP surfaces adsorb Ag+ ions in silver colloidal solutions. Our final solution contained three species of silver: Ag0 (AgNPs), Ag+ (silver ions), and Ag0/Ag+ (Ag+ adsorbed on Ag0) [2,95]. The presence of AgNPs was documented by transmission electron microscopy (TEM) imaging [95]. AgNPs were formed by transferring an extra anionic charge, producing positively charged AgNPs. Importantly, AgNPs show size-dependent effects [4], specifically smaller size AgNPs possess a high surface area relative to their total mass that increases the chance to interact with the parasite’s cells, and can easily enter the cytoplasm compared to larger AgNPs. The maximum concentration of free Ag+ has been detected in AgNPs with the highest surface area [96], and recent reports suggest that the free Ag+ cation is the most cytotoxic [97]. However, a previous study at the University of Texas San Antonio (UTSA) tested AgNPs (10.6 ng/L) and silver nitrate (silver ions 0.4 mg/L) in a ciliated parasite, and the results suggested a similar mode of action, but different toxicity, where silver ions were less toxic [98]. Therefore different effects may be observed depending on the size and composition of the three species of silver and the sensitivity of the exposed organism [97]. In the following paragraphs, we discuss differences in gene expression and biological processes affected in *Cichlidogyrus* parasites exposed to our formulation of AgNPs.

### 3.1. AgNPs Disrupt Tegument Organization

The tegument is the epidermis of monogeneans; it has several functions, such as absorption and secretion of substances, osmoregulation, mechanical support, and protection against xenobiotics [99]. The parasite’s tegument is the first contact site for AgNPs, which cause damage at the ultrastructural level [12]. AgNPs activated genes acting on the epidermis such as *cut-1*, *nhr-23*, *ifb-1*, *pat-6*, *ajm-1* and *lon-3.* Parasites exposed to the lower concentration of AgNPs showed up-regulation of the intermediate filament protein (*ifb-1*) gene, which is part of the cytoskeleton in many metazoan cells, providing mechanical resistance [100]. In *C. elegans*, *ifb-1* performs an important role in the transmission of muscle force to the cuticle and to the maintenance of the correct hypodermis/muscle relationship; lack of *ifb-1* gene causes morphological defects and defective excretory cells [96]. At the same time, parasites exposed to the higher concentration of AgNPs showed up-regulation of genes acting in body wall muscle attachments, such as *pat-6*/actopaxin [69,101] and *ajm-1* (apical junction component 1) located on the apical junctional domain of the basal epithelia of *C. elegans* to the HMR–HMP (cadherin–catenin) complex, which is important for the structural maintenance of the epithelium [71,102], and *lon-3*, secreted by the hypodermis and required for body size regulation [88,103]. At both AgNP concentrations, the gene cuticlin-1 (*cut-1*) was down-regulated; this gene encodes a non-collagenous component of *C. elegans* cuticle that contributes to the formation of extracellular envelopes protecting the organism from the environment [104]. Lack of *cut-1* causes an increase in diameter, reduction in size and loss of the rings for dauer larvae [58]. Thus, exposure of the parasites to AgNPs affected tegument organization; the low concentration caused problems associated with force and mechanical resistance, whereas the high concentration affected structural integrity. 

### 3.2. AgNPs Activate Detoxification Mechanisms

Exposure to the lower concentration of AgNPs induced the expression of a multidrug resistance protein, *mrp-4*, a member of the C subfamily of ATP-binding cassette (ABC); these proteins are mainly located in the cell membrane in organs such as the tegument, intestine, brain capillaries, and liver. *mrp-4* pumps xenobiotics out of the cell, protecting it from cytotoxic compounds [105,106]. ABC transporters are presented in Platyhelminthes, with the ATP-binding cassette C (ABCC) subfamily well represented in monogenean parasites [107]. *mrp-4* up-regulation suggests that AgNPs enter through the tegument and activate cellular detoxification processes to expel nanoparticles outside the cells as a defense mechanism. However, the down-regulation of the nuclear hormone receptor gene, *nhr-49*, indicates a decreased detoxification response. In *C. elegans*, under normal conditions, *nhr-49* is essential for the expression of the glutathione-s-transferase gene, *gst-4*, which participates in phase II detoxification processes [62]. Therefore, decreased expression of *nhr-49* indicates that AgNPs are not completely expelled, and that some nanoparticles are able to pass through the epidermis. 

Exposure to a higher concentration of AgNPs affected a different detoxification mechanism, decreasing the expression of *pcs-1*, which encodes the enzyme phytochelatin synthase, related to heavy metal detoxification. *pcs-1* is expressed in the hypodermis, pharynx, and body wall muscles [82], and its expression increases tolerance to heavy metals [108]. In *C. elegans*, *pcs-1* plays an important role in cadmium detoxification; knockout of *pcs-1* causes developmental retardation and early death [81]. The detoxification pathway regulated by *pcs-1* is specific for some metals such as cadmium, contrary to the usual expression of metallothioneins [109]. Since metallothionein genes were not activated, in *Cichlidogyrus, pcs-1* down-regulation may indicate a more specific response to AgNPs. 

### 3.3. AgNPs Affect the Nervous System

AgNPs cause alterations in the nervous system of parasites affecting neuroarchitecture and neurotransmission. In parasites exposed to the low concentration, *zig-5* was upregulated (+4.75 z-score); in *C. elegans, zig* genes are necessary to maintain the correct position of neural soma and ventral cordon axons through the inhibition of *sax-7* [36]. Other genes showing differential expression in the parasites were *hmr-1*, *ahr-1*, *cfz-2*, and *zag-1*. *hmr-1* encodes for a cadherin that participates in the regulation of axonal patterns and epithelial morphogenesis in motor neurons [43,110]; *ahr-1*—which is orthologous of the mammalian aryl hydrocarbon receptor (AhR), a ligand-activated transcription factor that mediates the toxic effects of environmental pollutants [111]—is related to neuronal differentiation and development [112]. In *Cichlidogyrus* parasites, *ahr-1* was down-regulated probably due to the toxic effect of AgNPs on the neuronal anatomy. *cfz-2* seems to be related to neuronal organization in *C. elegans*; in studies with *cfz-2* mutants, defects in the development of neuronal axons have been observed and the organization of neurons is interrupted [54]. The Zn-finger–homeodomain *zag-1* gene showed strong down-regulation (−3.08 z-score); *zag-1* has a role in the differentiation of neuronal cells in *C. elegans*, and *zag-1* mutants show defects in axonal guidance [51]. This gene also intervenes in the expression of other factors like *ceh-28*, which activates expression of growth factor genes *dbl-1* and *egl-17*, and neuropeptide genes *flp-5* and *flp-2* [113]. The down-regulation of *zag-1* in *Cichlidogyrus* parasites suggests that AgNPs affect neuronal differentiation and neurotransmission. 

Neurotransmission needs the correct function of synaptic vesicles containing neurotransmitters released into the synaptic cleft that communicates to the axon or soma of other neurons. Once the neurotransmitters are released, the vesicle proteins recovered from the plasma membrane are recycled into new synaptic vesicles to maintain neuronal transmission [114]. AgNPs decreased the expression of genes involved in neuronal signaling in synaptic vesicle formation and recycling, such as *unc-26* (synaptojanin), *unc-11* (synaptobrevin) and *fat-3* (6-desaturase). In *C. elegans*, inactivation of *unc-26* causes decrease in synaptic vesicles; this gene is important for recycling and coating vesicles [66,112]. *unc-11* is expressed at the presynaptic terminals, and mutants show affectations in the biogenesis of synaptic vesicles due to the reduction in synaptobrevin, which decreases the release of neurotransmitters [55,56]. *fat-3* is essential in the biosynthesis of long-chain polyunsaturated fatty acids (LC-PUFA) for the formation of neural membranes. Worms lacking *fat-3* function do not synthesize LC-PUFAs and show movement and egg-laying abnormalities associated with neuronal impairment. Absence of *fat-3* in *C. elegans* reduces the amount of synaptic vesicles, producing low levels of neurotransmitters [60,61].

Exposure to xenobiotics can cause neurotoxic effects [115,116,117]. AgNP exposure can cause high neuronal activity, which needs the fusion of synaptic vesicles with the plasma membrane at high speed, and sustained vesicle fusion rates need an efficient synaptic vesicle recycling mechanism [118]. Low expression of genes related to formation and recycling of synaptic vesicles can alter the correct neurotransmission in *Cichlidogyrus* parasites, causing an imbalance in homeostatic processes. 

Parasites exposed to high concentrations of AgNPs also displayed effects on the nervous system. For instance, the odorant receptor gene *str-2* was up-regulated (4.79 z-score). *str-2* is related to the response to environmental stimuli by chemosensory neurons. It is expressed in a type of olfactory neurons (AWC) in volatile odor chemotaxis processes in *C. elegans*, and gene inactivation confers deficiency in chemical sensitivity [52]. Although AWC neurons have not been described in monogenean parasites, these parasites have sensory systems [119], and activation of *str-2* may indicate that AgNPs triggered a chemosensory response. These parasites also showed decreased expression of *zag-1*, affecting the correct functioning of neuromotor cells and neuropeptide secretion [113]. Other down-regulated genes were *ehs-1* and *ldb-1* genes; in *C. elegans,* a low activity of the *ehs-1* gene decreases the number of synaptic vesicles and causes uncoordinated movements due to presynaptic defects in neurotransmission [94], whereas inactivation of *ldb-1* causes motility defects, such as worm lethargy, probably due to loss of mechanosensory activity [93].

*Cichlidogyrus* parasites exposed to both concentrations of AgNPs showed lethargy and abnormal body contractions before dying; in toxicological studies, this motor response has been observed due to the neurological damage caused by toxins [120]. Akter et al. [121] mention that AgNPs can act as a neurotoxin, and several studies conducted with mice report effects such as inflammation, increased permeability, neuronal damage and degeneration in the synapse. The increase in ROS levels caused by AgNPs generates oxidative stress that can cause neurodegeneration in neuronal cells [122]. In addition, it has been demonstrated that the brain of monogenean parasites has affinity for silver compounds; for instance, El-Naggar et al. [123] reported affinity of the brain of two species of monogenean parasites with silver, therefore, silver from AgNPs could bind to neural structures and cause damage in the nervous system of parasites. Furthermore, it was shown that neurons are the second target organ of nanoparticles in *C. elegans* [124].

### 3.4. AgNPs Activate Signaling Pathways

Genes involved in cell signaling are activated to provide defense against damage caused by AgNPs. The ortholog gene of human MAP2K7 (mitogen-activated protein kinase 7), *jkk-1*, was up-regulated in parasites exposed to the higher concentration of AgNPs. *jkk-1* is a member of the MAP kinase superfamily and functions as a specific activator of JNK (c-Jun N-terminal kinase). In vertebrates, the JNK cascade can be activated as a mechanism of resistance to toxicity from heavy metals or other environmental stressors to extend the life span of organisms [75,79,125]. JNK acts as a positive regulator in insulin/DAF-16-dependent signaling to regulate longevity and stress resistance [126,127]. The *jkk-1* gene is expressed in most neurons in *C. elegans*, and its inactivation can cause defects in locomotion. In addition, the JNK pathway works in motor neurons, modulating coordination in locomotion [80,128]. *Cichlidogyrus* parasites exhibited lethargy in their movements after exposure to AgNPs, thus the expression of *jkk-1* could also indicate the effect of AgNPs on neurolocomotion. 

Interestingly, the *vhp-1* gene was up-regulated, and this gene acts negatively on JNK signaling. *vhp-1* encodes a mammalian homologous MPK7, which dephosphorylate p38 and JNK MAPKS. Loss of function of these pathways results in hypersensitivity of *C. elegans* to heavy metals [79,129]. 

AgNPs at both concentrations activated serine/threonine kinase genes, *akt-1* and *akt-2*. These signaling pathways are involved in several cellular processes, such as glucose metabolism (insulin/IGF-1), protein synthesis (mTOR), and cell proliferation and survival (FOXO) [130]. In *C. elegans*, *akt-1* and *akt-2* act in the insulin and insulin-like growth factor signaling (IIS) pathway to regulate lifespan, development, metabolism, and stress resistance [48]. It has also been observed in *C. elegans* that *akt-1* and *akt-2* regulate development in response to environmental conditions by controlling FOXO/DAF-16 [68]. *akt-1* and *akt-2* also act as anti-apoptotic proteins to modulate DNA damage-induced programmed cell death in *C. elegans* germ lines. This process has been observed in several species, so the ability of AKT to prevent apoptosis generated by DNA damage seems to be evolutionarily conserved [68]. *Cichlidogyrus* parasites may express these genes in response to cellular and/or DNA damage caused by AgNPs. Several studies have concluded that AgNPs can interact with DNA by breaking double strand bonds [34,131,132].

Parasites exposed to the lower concentration of AgNPs showed down-regulation of *pqm-1*, which is involved in cell signaling as a transcriptional activator for the control of genes associated with DAF-16. Loss of *pqm-1* suppresses DAF-2 activity, which can inactivate *daf-16* transcription, probably affecting the resistance of parasites to AgNP exposure as well as accelerated cell death. The decreased expression of *pqm-1* is related to aging [60,133]. Likewise, the phospholipase D gene *pdl-1* was up-regulated by a 3.51 z-score, and it is also involved in the DAF-16 pathway. Park et al. [40] observed that *pdl-1* down-regulation causes ROS accumulation and decreases longevity. Thus, the increased expression of *pdl-1* may indicate a damage response by *Cichlidogyrus* parasites.

### 3.5. AgNPs Cause DNA Damage and Cell Death

Exposure to low concentrations of AgNPs can activate genes involved in cell death. Cell death mechanisms are activated when organisms are unable to repair damaged cells [134]. The down-regulation of *hsr-9*, homolog of the mammalian 53BP1 (p53 binding protein 1), suggests that AgNPs at low concentrations may damage the DNA in *Cichlidogyrus* parasites; *hsr-9* has a role in cell cycle checkpoints, DNA repair and apoptosis. In *C. elegans hsr-9* is involved in DNA repair by non-homologous end-joining (NHEJ) and suppresses homologous recombination (HR) [67]. DNA damage induced by AgNPs has been well defined [24,26], thus, while the expression of *hsr-9* may indicate DNA repair, low expression levels of *hsr-9* may indicate low efficiency in DNA repair, triggering cell death processes.

Parasites exposed to low concentrations of AgNPs also showed up-regulation of *pqn-41* (4.15 z-score); this gene is involved in programmed cell death through a non-apoptotic route independent of caspases [39]. This process has been observed in linker cells during the reproductive development of *C. elegans* males [135]. In vertebrates, this type of non-apoptotic cell death has been observed in the normal development of neurons in the spinal cord and ciliary ganglia [136]. In addition, patients with neurodegenerative disease associated with polyglutamine like *pqn-41* show crenellated nuclei and swollen endoplasmic reticulum and mitochondria in dying neurons [39,135]. Neurodegeneration has been observed in nematodes exposed to nanoparticles [135]; thus, the expression of *pqn-41* in *Cichlidogyrus* parasites could be related to the damage caused by AgNPs in the nervous system. 

### 3.6. AgNPs Affect Reproduction and Embryonic Development

Exposure to both concentrations of AgNPs activated genes that regulate reproduction and embryonic development; this response may be a survival strategy for parasites. Increased reproduction, oviposition, and embryonic development have been documented as a response of nematode parasites to xenobiotics [137]. In some organisms, the response to environmental toxins involves reproductive changes; for instance, in hydra species, exposure to toxins induces sexual reproduction [122,138]. Some studies have shown that AgNPs cause adverse reproductive effects on non-mammalian model organisms [138].

*Cichlidogyrus* parasites exposed to 6 µg/L of AgNPs over-expressed genes related to sperm activation (*spe-29*, *rrf-3*), embryonic development (*spe-11*, *cyb-2.1*, *hmr-1*), and gonadal differentiation (*lin-9*). Specifically, the *lin-9* gene was highly up-regulated (4.67 z-score). In *C. elegans*, *lin-9* is necessary for the development of the hermaphrodite gonads and the male reproductive system, spicule, rays, and gonads [37,139]. Similarly, *syp-3* regulates synapses along chromosomes, contributing to meiotic progression in early prophase during reproduction [41]. Higher concentrations of AgNPs (36 µg/L) also activated genes involved in oviposition and embryonic development in parasites (*pal-1*, *goa-1*, *cdc-25.1*, and *ptl-1*). *pal-1* is necessary to maintain the development of the multipotent C blastomere lineage in the *C. elegans* embryo [69]; interestingly, *tra-2* was up-regulated in *Cichlidogyrus* parasites, and this gene is involved in sexual determination in *C. elegans* [81]. Reproductive alterations have been observed after exposure to a variety of nanoparticles in the nematode *C. elegans*, decreasing reproductive capabilities [122,140].

### 3.7. Final Remarks

A regulatory process depends on the activation and silencing of different genes in a cell/tissue and/or time-specific fashion. Alteration of gene expression under toxic exposure indicates toxicity-mediated dysregulation of a given process, which, in turn, affects other processes in a regulatory network, compromising organ/body function, and ultimately producing disease or death. As a (molecular) mechanistic analysis implies that a complex system (such a regulatory network) can be studied and understood by the examination of individual genes, we selected top up- and down-regulated genes to understand the effects of exposure of *Cichlidogyrus* parasites to high and low concentrations of AgNPs at one point time (1 h of exposure), established by a previous kinetic analysis [12] in which death was observed at the higher concentration (36 μg/L AgNPs) but not at the lower concentration (6 μg/L AgNPs). 

Somewhat unexpectedly, the same processes were affected in parasites exposed to low and high concentrations of AgNPs, but showing altered expression of different sets of genes (Figure 4). It has been suggested that gene expression after exposure to toxic compounds occurs by stages [141]: in the first stage, adaptation, which occurs after exposure to low concentrations of toxic chemicals, the organisms activate metabolic enzymes to detoxify and maintain homeostasis. In the parasites, genes involved in detoxification processes were observed at the lower concentration, in which a multidrug-resistance mechanism was activated. Then, the compensatory phase after exposure to higher concentrations activates DNA repair and programmed cell death mechanisms in order to survive—these processes were activated mostly at the lower concentration tested. Finally, exposure to high concentrations conduces to exhaustion and triggers the activation of adversity genes (including genes involved in accidental cell death and inflammation), shutting down cell survival and DNA repair processes, dismantling cellular components and compromising integrity. In the parasites, detoxification mechanisms were inactivated at the higher concentration and, in addition, different signaling pathways—known to be affected in response to heavy metals and related to neurological function (involving neurotransmission and neuroarchitecture), cell proliferation and survival, as well as tegument organization and integrity—were mostly altered at this stage. Importantly, each stage overlaps with the next, in such a way that adaptation–compensation genes, and compensation–adversity genes could be activated or silenced at different points during exposure. According to Pimentel-Acosta et al. [12], the tegument was affected at the ultrastructural level in parasites exposed to both concentrations of AgNPs: at 6 μg/L, the tegument showed vacuoles and slight swelling, suggesting a compensatory process, whereas at 36 μg/L, the tegument presented swelling, loss of corrugations, and disruption, indicating a more severe effect involving a different set of (adversity) genes. Not less important, genes involved in reproduction and development were probably the most sensitive to the exposure to both concentrations of AgNPs, and probably this response at a molecular level was related to a tradeoff of reproduction for survival, as it has been documented and discussed elsewhere [142].

Taken together, the results suggest that parasites exposed to 6 μg/L of AgNPs expressed a set of genes involved in adaptation and compensatory processes in an attempt to survive and reestablish homeostasis, whereas parasites exposed to 36 μg/L of AgNPs expressed a set of genes involved in compensatory and adversity processes; unable to restore homeostasis, this concentration was lethal within 1 h of exposure, as previously shown [12]. Thus, biological processes overlapped at both concentrations but were driven by different sets of genes depending on the ability of the organisms to cope with toxicity. 

In order to better understand toxicological effects of AgNPs, chemical–biological interactions should be assessed in future works at the cellular/molecular level by a quantitative structure–activity relationship (QSAR) study, according to Putz and Dudaş [143,144]. That is, the structure of a toxic compound could be related to its activity, functionality, and toxicology by computational modeling of chemical–biological interactions with the cells. By this approach, it would be possible to establish the correct toxicological dose and to model the toxic effects of AgNPs in both parasite and host. This kind of mechanistic approach is not trivial, and requires the identification of interactions at the chemical–biological scale, in such a way that the structural properties of a compound (in this case AgNPs) can be related to the effects in a biological system (in this case the molecular–cellular interactions within the parasite). This is an exciting line of research that will help in answering several questions regarding mechanisms of AgNP toxicity and, at the same time, will most likely open new perspectives in this field. 

## 4. Materials and Methods

### 4.1. Silver Nanoparticles

Silver nanoparticles were a gift from the Department of Physics of the University of Texas at San Antonio. AgNPs are well characterized as previously described [2,95]. Nanoparticles were measured by transmission electron microscopy (TEM) and the average size distribution was 1–3 nm with round morphology. This synthesis resulted in pure silver nanoparticles without the adding of other chemical reagents, as the synthesis route was physical (microwave-assisted route) in solution. The surface potential of the solution was studied by the zeta potential, and the value augmented from negative (−2.9 mV) to positive values (+13.4 mV) over a time period of 120 h. This change to positive charge shows the adsorption of Ag+ cations from the solution on the metallic nanoparticles. The concentration of AgNPs was 0.016 mg/mL of metallic silver. Dilutions were made with distilled water to obtain working concentrations of 6 and 36 mg/L of silver.

### 4.2. Monogenean Parasites

Monogenean parasites of the genus *Cichlidogyrus* were collected from the gills of Nile tilapia obtained from a freshwater fish farm in Southern Sinaloa, Mexico. Parasites were pooled together because the identification of some species is time consuming due to their morphological similarity. 

### 4.3. Experimental Design and Sample Collection

In a previous study, anthelmintic concentrations of AgNPs were obtained and studied to eliminate *Cichlidogyrus* parasites, and the ultrastructural lesions were observed by SEM electron microscopy [12]. Based on these results, two concentrations were selected, a low concentration (6 µg/L Ag) and a high concentration (36 µg/L Ag). Approximately 1800 parasites per treatment (three treatments consisting of control, low and high Ag concentrations) were exposed for 1 h in vitro according to Pimentel-Acosta et al. [12]. Live parasites were collected and fixed in RNAlater solution and stored at 4 °C for few days until processing.

### 4.4. RNA Extraction

Total RNA was extracted using RNeasy^®^ Plus Micro Kit (QIAGEN, Aspelab, Mexico City, Mexico) according to the manufacturer’s protocol. RNA quantity was determined using a NanoDrop DeNovix^®^ DS-11 spectrophotometer, and the quality was analyzed by 1% agarose gel electrophoresis. 

### 4.5. Heterologous Microarray Analysis

Genomic information of *Cichlidogyrus* parasites is scarce, thus, in order to study the mechanisms of action of AgNPs at the molecular level, we used a heterologous microarray approach, using a *C. elegans* microarray.

#### 4.5.1. Printing of Arrays

The *C. elegans* 70-mer oligo library from OPERON Oligo Sets (http://omad.operon.com/) was used. It contains 20,000 gene-specific oligonucleotide probes representing the complete genome. Oligo library was resuspended to 40 µM in Micro Spotting solution (ArrayIt Brand Products, Probiotek, Mexico City, Mexico). SuperAmine coated slides 25 × 75 mm (TeleChem International Inc, Sunnyvale CA, USA) were printed and fixed at 80 °C for 4 h. Slides were re-hydrated with water vapor at 60 °C, fixed with two cycles of UV light (1200 J), heated for two minutes at 92 °C, washed with 95% ethanol for one minute, and prehybridized in 5X sodium chloride, sodium citrate buffer (SSC), 0.1% sodium dodecyl sulfate (SDS) and 1% bovine serum albumin (BSA) for one hour at 42 °C. Then, the slides were washed and dried for further hybridization.

#### 4.5.2. Probe Preparation and Hybridization to Arrays

An amount of 5 µg of total RNA was used for cDNA synthesis, incorporating dUTP-Alexa555 or dUTP-Alexa647 and employing the First-Strand cDNA labeling kit (Invitrogen, Thermo Fisher Scientific, Mexico City, Mexico). Incorporation of fluorophore was analyzed by using the absorbance at 555 nm for Alexa555 and 650 nm for Alexa647. Equal quantities of labeled cDNA were hybridized using the UniHyb (TeleChem International Inc) hybridization solution on the 22,000 *C. elegans* oligo arrays for 14 h at 42 °C. The microarrays were hybridized for three conditions: untreated (control) parasites, parasites treated with 6 µg/L AgNPs, and parasites treated with 36 µg/L AgNPs. 

#### 4.5.3. Data Acquisition and Analysis of Array Images

Acquisition and quantification of array images was performed in a GenePix 4100-A reader with its accompanying GenePix from Molecular Devices software. For each spot, the Alexa555 and Alexa647 density mean value, background mean value, and signal cross channel lower normalization value (by subgrids) were calculated with the ArrayPro Analyzer software from Media Cybernetics (Rockville, MD, USA).

### 4.6. Data Analysis

Microarray data analysis was performed with the free software genArise; this software was developed in the Computing Unit at the Institute of Cellular Physiology (UNAM) (http://www.ifc.unam.mx/genarise/). GenArise performs a number of transformations, such as background correction; it lowers normalization, intensifies the filter, replicates the analysis, and ultimately selects differentially expressed genes. Selection of differentially expressed genes is achieved by calculating an intensity-dependent z-score, using a sliding window algorithm to calculate the mean and standard deviation within a window surrounding each data point. It defines a z-score, where z measures the number of standard deviations a data point is away from the mean.
zi = (Ri − mean(R))/sd(R)(1)
where zi is the z-score for each element, Ri is the log ratio for each element, and sd(R) is the standard deviation of the log ratio. With this criterion, the elements with a z-score > 2 standard deviations were defined as significantly differentially expressed genes.

The analysis of the results was performed with the Bioinformatics Resources DAVID 6.8 to identify molecular pathways affected by AgNP exposure in monogenean parasites by differentially expressed genes. A ≥2 z-score (*p* ≤ 0.05) boundary in gene expression was set to identify up- and down-regulated genes.

## 5. Conclusions

The heterologous microarray hybridization approach used in this study allowed the identification of biological processes affected by exposure to AgNPs in monogenean parasites. Exposure to low and high concentrations of AgNPs activated genes involved in detoxification, neurotoxicity, cell signaling, cell death, DNA damage, reproduction, and embryonic development. Both concentrations of AgNPs induced similar biological processes in *Cichlidogyrus* parasites through different mechanisms.

## Figures and Tables

**Figure 1 ijms-21-05889-f001:**
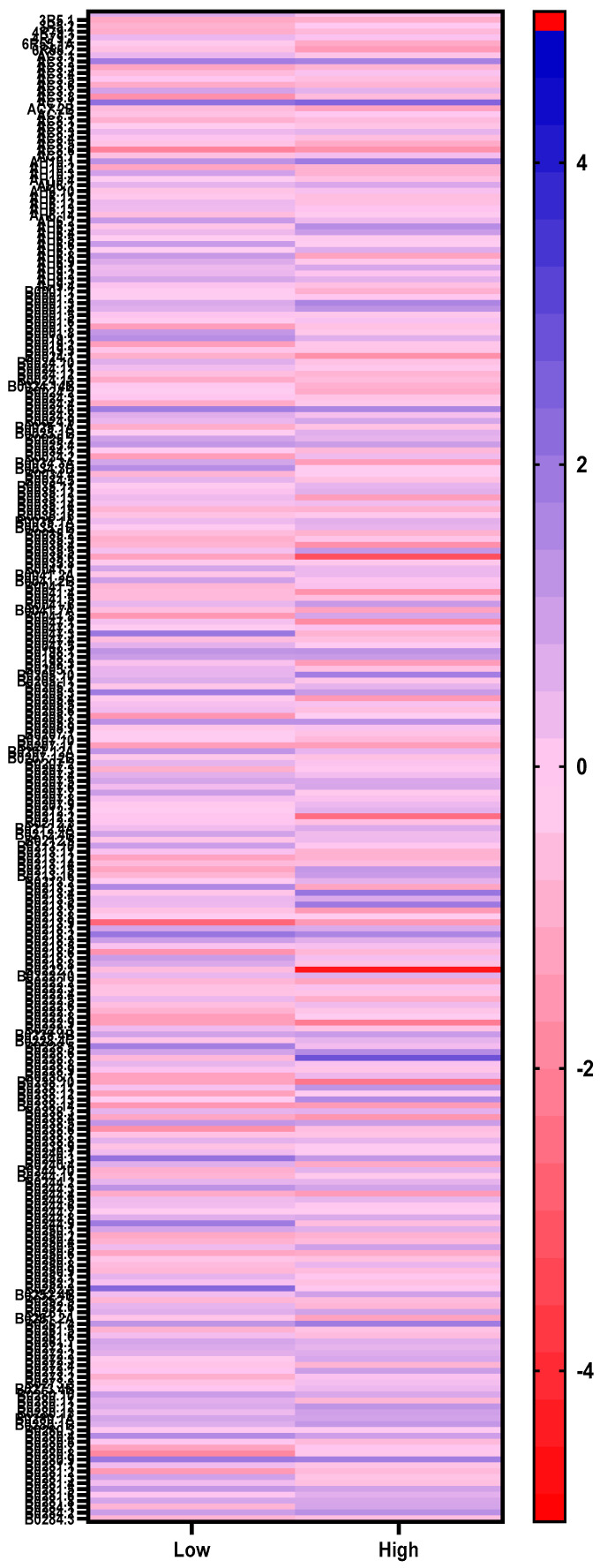
Heatmap showing differentially expressed genes in parasites exposed to low (6 µg/L) and high (36 µg/L) concentrations of silver nanoparticles.

**Figure 2 ijms-21-05889-f002:**
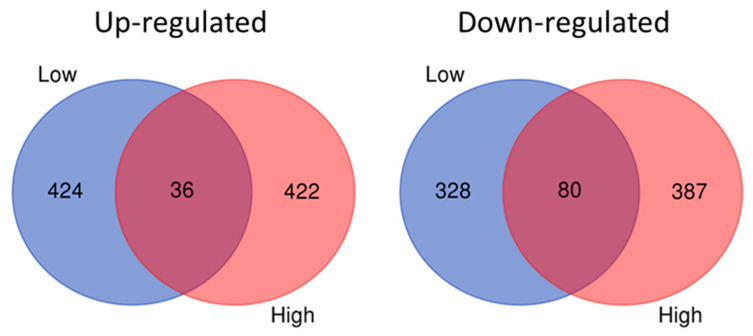
Venn diagrams showing up- and down-regulated genes in parasites exposed to low (6 µg/L) and high (36 µg/L) concentrations of silver nanoparticles.

**Figure 3 ijms-21-05889-f003:**
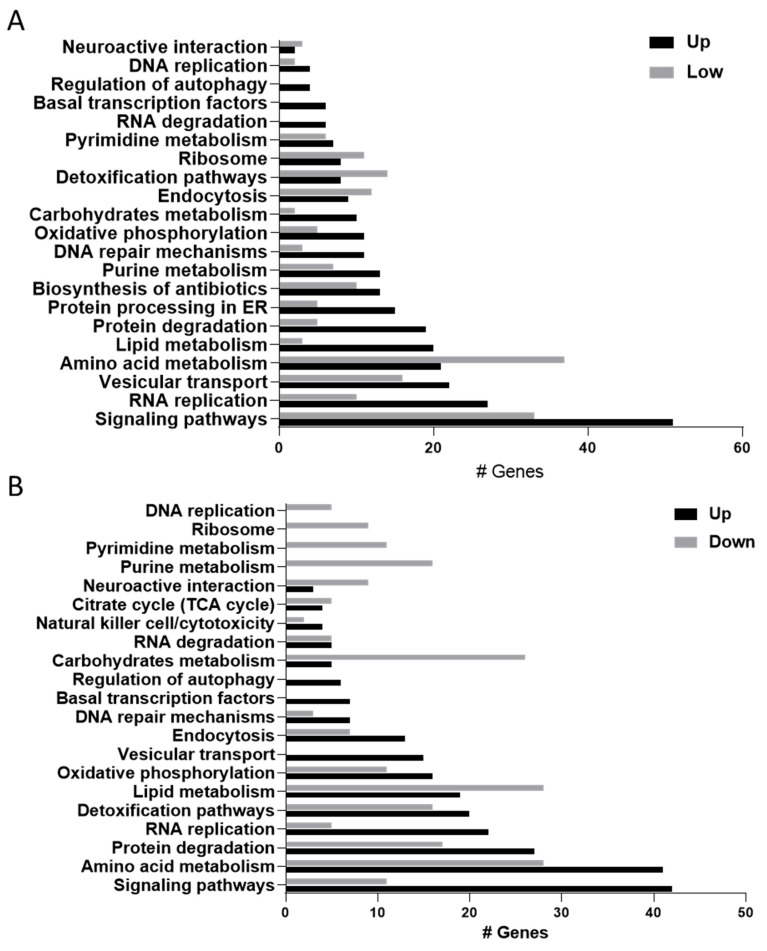
Molecular pathways activated in *Cichlidogyrus* parasites exposed to silver nanoparticles (AgNPs). (**A**) Low concentration (6 µg/L), (**B**) high concentration (36 µg/L). Black bars represent up-regulated genes, while grey bars represent down-regulated genes in each experimental condition.

**Figure 4 ijms-21-05889-f004:**
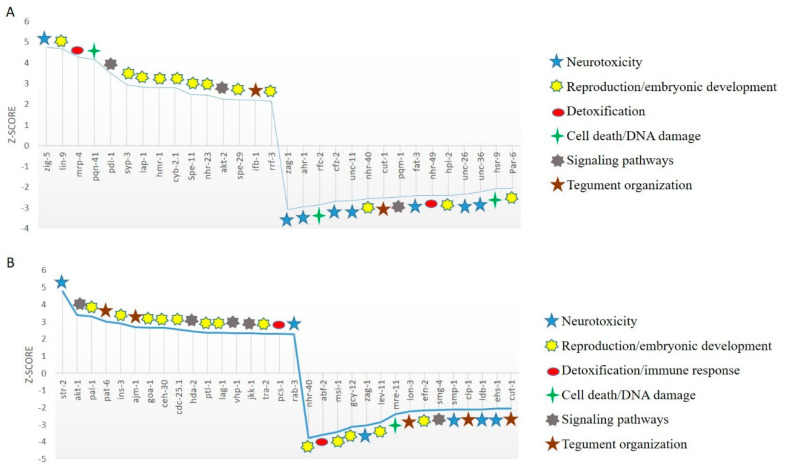
Top up- and down-regulated genes (z-scores) in *Cichlidogyrus* parasites exposed to (**A**) low (6 µg/L) and (**B**) high concentrations (36 µg/L) of AgNPs.

**Table 1 ijms-21-05889-t001:** Top 30 differentially expressed genes in *Cichlidogyrus* parasites exposed to 6 µg/L AgNPs for 1 h (+ upregulated, − downregulated).

ID	Symbol	Fold Change	Function	Reference
Y48A6A.1	zig-5	+4.75	Neural cell adhesion	[36]
ZK637.7B	lin-9	+4.67	Gonadal differentiation, formation of sexual organs	[37]
F21G4.2	mrp-4	+4.28	Detoxification, membrane transporters via lysosomes.	[38]
F53A3.4	pqn-41	+4.15	Programmed cell death	[39]
C27H5.1	pdl-1	+3.51	Signaling pathways, forkhead box O (FOXO)/abnormal dauer formation (DAF-16)	[40]
F39H2.4	syp-3	+2.93	Meiosis, chromosome segregation	[41]
ZK353.6	lap-1	+2.83	Larval development, cuticle degradation, digestive enzyme	[42]
W02B9.1B	hmr-1	+2.81	Embryonic development	[43]
Y43E12A.1	cyb-2.1	+2.8	Embryonic development	[44]
F48C1.7	Spe-11	+2.47	Embryonic development	[45]
C01H6.5B	nhr-23	+2.45	Larval development, molting	[46]
F28H6.1B	akt-2	+2.23	Signaling pathways, FOXO/DAF-16 Antiapoptotic activity	[47]
F25H8.7	spe-29	+2.21	Reproduction: sperm activation	[48]
F10C1.2B	ifb-1	+2.19	Muscular junction to cuticle	[49]
F10B5.7	rrf-3	+2.14	Spermatogenesis	[50]
F28F9.1	zag-1	−3.08	Neuronal differentiation	[51]
C41G7.5	ahr-1	−2.96	Neurotransmission	[52]
F58F6.4	rfc-2	−2.87	Embryogenesis DNA repair by nucleotide cleavage	[53]
F27E11.3A	cfz-2	−2.67	Neuronal organization	[54]
C32E8.10C	unc-11	−2.66	Synaptic vesicle biogenesis	[55,56]
T03G6.2B	nhr-40	−2.57	Morphogenesis, development of muscle cells in epithelium	[57]
C47G2.1	cut-1	−2.53	Cuticle formation	[58]
F40F8.7	pqm-1	−2.48	pqm-1 loss suppresses daf-2 expression and slows development	[59]
W08D2.4	fat-3	−2.42	Synaptic vesicle biogenesis and recycling	[60,61]
K10C3.6A	nhr-49	−2.42	Detoxification phase II	[62]
K01G5.2A	hpl-2	−2.41	Stress resistance	[63]
JC8.10B	unc-26	−2.35	Recycling of synaptic vesicles	[64]
C50C3.9	unc-36	−2.24	Functional activity of the mechanosensory neurons	[65]
T05F1.6A	hsr-9	−2.06	Apoptosis, DNA repair	[66]
T26E3.3	Par-6	−2.06	Embryogenesis	[48]

**Table 2 ijms-21-05889-t002:** Top 30 differentially expressed genes in *Cichlidogyrus* parasites exposed to 36 µg/L AgNPs for 1 h (+ upregulated, − downregulated).

Id	Symbol	Fold Change	Function	Reference
C50C10.7	str-2	+4.79	Neural development	[52]
C12D8.10B	akt-1	+3.38	Signaling pathways, FOXO/DAF-16 Antiapoptotic activity	[67]
C38D4.6	pal-1	+3.31	Embryonic development	[68]
T21D12.4	pat-6	+3.02	Muscle cell adhesion during maturation	[69]
ZK75.3	ins-3	+2.9	Stress resistance	[70]
C25A11.4A	ajm-1	+2.68	Apical junction of epithelia	[71]
C26C6.2	goa-1	+2.65	Embryonic development	[72]
C33D12.7	ceh-30	+2.65	Synaptic vesicle biogenesis	[73]
K06A5.7	cdc-25.1	+2.55	Embryonic development	[74]
C08B11.2	hda-2	+2.45	Organization and remodeling of chromatin	[75]
F42G9.9D	ptl-1	+2.37	Embryonic development and mechanosensory neurons	[76]
K08B4.1A	lag-1	+2.36	Transcriptional regulator, signaling in nematode development	[77]
F08B1.1A	vhp-1	+2.34	Negatively regulates the mitogen-activated protein kinase (MAPK)-Jun N-terminal kinase (JNK) pathway. Negative regulator of stress tolerance	[78]
F35C8.3	jkk-1	+2.33	Active pathway JNK	[79]
C15F1.3B	tra-2	+2.31	Sexual determination in males	[80]
F54D5.1	pcs-1	+2.3	Heavy metal detoxification	[81,82]
C18A3.6A	rab-3	+2.28	Regulation of vesicles in endocytosis and exocytosis	[55]
T03G6.2B	nhr-40	−3.76	Morphogenesis, development of muscle cells in epithelium	[57]
C50F2.10	abf-2	−3.57	Immune response	[83]
R10E9.1	msi-1	−3.43	Reproduction	[84]
F08B1.2	gcy-12	−3.11	Body size regulation	[85]
F28F9.1	zag-1	−3.03	Neural development	[51]
Y105E8B.1D	lev-11	−2.85	Sexual differentiation	[86]
ZC302.1	mre-11	−2.39	DNA repair	[87]
ZK836.1	lon-3	−2.22	Cuticle formation	[88]
C43F9.8	efn-2	−2.16	Embryonic development	[89]
F46B6.3B	smg-4	−2.15	Corrects aberrant transcripts	[90]
Y54E5B.1B	smp-1	−2.12	Embryogenesis	[91]
C06G4.2B	clp-1	−2.11	Muscular dystrophy	[92]
F58A3.1C	ldb-1	−2.1	Neural development	[93]
ZK1248.3A	ehs-1	−2.07	Synaptic vesicle recycling	[94]
C47G2.1	cut-1	−2.06	Formation of longitudinal ridges in cuticle	[58]

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
