# Peer review of "Molecular Effects of Silver Nanoparticles on Monogenean Parasites: Lessons from *Caenorhabditis elegans"

_ijms, 2020, doi:10.3390/ijms21165889_

Round 1
Reviewer 1 Report
This paper described a study in which monogenean parasites were exposed to two different concentrations of silver nanoparticles and subsequent transcriptomics changes were followed. Presentation of the results is clear and results are well explained.
Some concerns relate however with the selection of nanoparticles. It was mentioned that the size of Ag NPs was 1-3 nm but no proof was given (e.g., TEM image), also the size of Ag NPs in exposure media was not reported. How stable were AgNP suspensions and which size were the particles in exposures?
The latter comment brings me to my second point which is the role of Ag ions in the observed results. Do the authors think that the observed changes in gene regulation were due to nanoparticles or released silver/ions? In this work, Ag ions were not included for comparison. Have the authors seen any studies on C.elegans where a transcriptomic response to Ag ions was followed? I would suggest discussing the role of nanoparticulates and silver ions in the toxicity of Ag NPs the Introduction, if possible.
Finally, it is interesting that although the molecular pathways activated or repressed in Cichlidogyrus by the low and high Ag were quite similar, the affected genes themselves were quite different. Do the authors have an explanation for that and could this be discussed in the manuscript more thoroughly?
Author Response
We appreciate the comments from the Editor and the Reviewers, which we believe improved the contents of this manuscript.
Answers to Reviewer 1
This paper described a study in which monogenean parasites were exposed to two different concentrations of silver nanoparticles and subsequent transcriptomics changes were followed. Presentation of the results is clear and results are well explained.
Some concerns relate however with the selection of nanoparticles. It was mentioned that the size of Ag NPs was 1-3 nm but no proof was given (e.g., TEM image), also the size of Ag NPs in exposure media was not reported. How stable were AgNP suspensions and which size were the particles in exposures?
This is an important point. The AgNPs were well characterized. Nanoparticles were measured by transmission electron microscopy and the average size distribution was 1-3 nm with round morphology. This synthesis resulted in pure silver nanoparticles without the adding of other chemical reagents, as the synthesis route was physical (microwave-assisted route) in solution. The surface potential of the solution was studied by the zeta potential, the Z potential value augmented from negative (−2.9 mV) to positive values (+13.4 mV) over a time period of 120 h. This change to positive charge shows the adsorption of Ag+ cations from the solution on the metallic nanoparticles. This has been added to the manuscript, in the Materials and Methods, section 4.1, lines 381-388.
The latter comment brings me to my second point which is the role of Ag ions in the observed results. Do the authors think that the observed changes in gene regulation were due to nanoparticles or released silver/ions? In this work, Ag ions were not included for comparison. Have the authors seen any studies on C.elegans where a transcriptomic response to Ag ions was followed?
To explain if the efficacy against parasites is due to silver ions or AgNPs, previous studies demonstrated that AgNPs surfaces adsorb Ag+ ions in silver colloidal solutions. Our final solution contained three species of silver; Ag0 (AgNPs), Ag+ (silver ions) and Ag0/Ag+ (Ag+ adsorbed on Ag0). The presence of the formed AgNPs was documented by TEM. AgNPs have size-dependent effects; specifically smaller AgNPs possess high surface area relative to their total mass that increases the chance to interact with the parasite’s cells and can easily enter the cytoplasm compared to larger AgNPs. The maximum concentration of free Ag+ was detected in AgNPs with the highest surface area. Recent reports suggest that the free Ag+ cation is the most cytotoxic, however, a previous study tested AgNPs UTSA (10.6 ng/L) and silver nitrate (silver ions 0.4 mg/L) in a ciliated parasite, and the results suggested a similar mode of action, but different toxicity, where silver ions were less toxic. Therefore different effects may be observed depending on the size and composition of the three species of silver and the sensitivity of the exposed organism. We have added this information to the Discussion section, lines 136-151.
I would suggest discussing the role of nanoparticulates and silver ions in the toxicity of Ag NPs the Introduction, if possible.
We have also added this paragraph to the Introduction section, lines 63-68:
“…some authors have suggested that the effects may be due to the synergy of AgNPs and the released silver ions and cause cytotoxicity in a dose-dependent manner [32]. In C. elegans some transcriptomic effects of AgNPs were similar to silver ions (AgNO3), indicating that both can alter gene expression profiles in pathways associated with stress response, reproduction an lysosomal activity; these results suggest that the effects are partially due to dissolved silver ions [32,35].”
Finally, it is interesting that although the molecular pathways activated or repressed in Cichlidogyrus by the low and high Ag were quite similar, the affected genes themselves were quite different. Do the authors have an explanation for that and could this be discussed in the manuscript more thoroughly?
We thank the reviewer for this recommendation. We used information regarding gene expression after exposure to environmental toxicants, in which the authors classified genes in three categories: adaptation, compensation, and adversity genes (depending on exposure time or toxicant concentration). We integrated these scenarios to our study, and added a “Final Remarks” section at the end of the Discussion, lines 337-380, plus an additional figure (Fig. 4) to help explain these differences.
Reviewer 2 Report
The paper showcases more on literature part than on original work done by authors. As it is now the paper seems a Review merely. I do strongly recommend authors re-distilling the original part by the literature one by improving the manuscript while presenting their vs. literature results, while clearly describing each step of the two distinguished mechanism in high/low concentration AgNPs in monogean parasites, in all stages identified, such as: activating genes involved in detoxification, neurotoxicity, cell signaling, cell death, DNA damage, reproduction, and embryonic development. I do recommend usage of kinetic analysis and structure-activity mechanistic approach for data analysis and comparative discussion, while best describing the ligand-receptor specific interactions. I will be happy to review the revised manuscript!
Author Response
We appreciate the comments from the Editor and the Reviewers, which we believe improved the contents of this manuscript.
Answers to Reviewer 2
The paper showcases more on literature part than on original work done by authors. As it is now the paper seems a Review merely. I do strongly recommend authors re-distilling the original part by the literature one by improving the manuscript while presenting their vs. literature results, while clearly describing each step of the two distinguished mechanism in high/low concentration AgNPs in monogean parasites, in all stages identified, such as: activating genes involved in detoxification, neurotoxicity, cell signaling, cell death, DNA damage, reproduction, and embryonic development. I do recommend usage of kinetic analysis and structure-activity mechanistic approach for data analysis and comparative discussion, while best describing the ligand-receptor specific interactions. I will be happy to review the revised manuscript!
We very much appreciate the Reviewer’s recommendation; there is not easy answer, in order to make sense of our results, we decided to use information regarding gene expression after exposure to environmental toxicants, in which the authors classified genes in three categories: adaptation, compensation, and adversity genes (depending, in this case, on toxicant concentration). We integrated these scenarios to our study, and added a “Final Remarks” section at the end of the Discussion, lines 337-380, plus an additional figure (Fig. 4) to help explain our results. We are not sure if we completely answered the question, but we found this final discussion very interesting, and we believe the article improved with it.
Regarding ligand-receptor-specific interactions, our formulation of AgNPs has size-dependent effects, but not ligand-receptor specificity; smaller AgNPs possess high surface area relative to their total mass that increases the chance to interact with the parasite’s cells and can easily enter the cytoplasm compared to larger AgNPs. This information was added to the Discussion section, lines 136-151.
Round 2
Reviewer 2 Report
Authors struggles to fulfill the tasks of the first revision round; the final remark section addition its thus appreciated; as well as the authors' response which indeed is sound towards a normal creative incompleteness of any research; however, as the final recommendation, minor this time, is to accommodate in their final remarks the reference with the mechanistic bonding algorithm as exposed on Structural Chemistry 24 (6), 1873-1893; Molecules 18 (8), 9061-9116 which may be further transferred for appropriate binding analysis at cellular level for the present case too. With the final recommendation implemented the paper may undergo acceptance.
Author Response
Authors struggles to fulfill the tasks of the first revision round; the final remark section addition its thus appreciated; as well as the authors' response which indeed is sound towards a normal creative incompleteness of any research; however, as the final recommendation, minor this time, is to accommodate in their final remarks the reference with the mechanistic bonding algorithm as exposed on Structural Chemistry 24 (6), 1873-1893; Molecules 18 (8), 9061-9116 which may be further transferred for appropriate binding analysis at cellular level for the present case too. With the final recommendation implemented the paper may undergo acceptance.
We appreciate the advice and the references (we indeed struggled to fulfill the tasks). We have revised both references and wrote the following paragraph at the end of the Discussion:
“In order to better understand toxicological effects of AgNPs, chemical-biological interactions should be, in future works, assessed at the cellular/molecular level by a quantitative structure-activity relationship (QSAR) study, according to Putz and Dudaş [146,147]. In other words, the structure of a toxic compound could be related to its activity, functionality, and toxicology by computational modeling of chemical-biological interactions with the cells. By this approach, it would be possible to establish a correct toxicological dose and to model the toxic effects of AgNPs in both parasite and host. This kind of mechanistic approach is not trivial, and requires the identification of interactions at the chemical-biological scale, in such a way that the structural properties of a compound (in this case AgNPs) can be related to the effects in a biological system (in this case the molecular-cellular interactions within the parasite). This is an exciting line of research that will help answering several questions regarding mechanisms of AgNPs toxicity, and, at the same time, will most likely open new perspectives in this field.”
We have revised the English language, found some minor mistakes and corrected them.